# Convergence or dispersion? The impact of China's innovative city pilot policy on regional innovation differences

**Yang Haodong, Wang Chen, Wang Gaofeng***

School of Humanities and Social Sciences, University of Science and Technology of China, Hefei, China

* wanggf@ustc.edu.cn

## Abstract

Although constructing innovative cities stimulates innovation, it may further widen regional innovation differences. Based on panel data from 275 cities in China from 2003 to 2020, the difference-in-differences method was used to examine the impact of the innovative city pilot policy on urban innovation convergence. The study finds that the pilot policy not only improves the innovation level of cities (basic effect) but also promotes innovation convergence among pilot cities (convergence effect). However, in the short-term, the policy slows the innovation convergence of the entire region. The results reveal the innovative city policy's multiple effects and dual character and capture the spatial spillover and regional heterogeneity of policy impact, highlighting the risk of further marginalizing some cities. This study supplements the evidence that government intervention affects regional innovation patterns based on the place-based innovation policy in China, providing theoretical support for expanding the follow-up pilot scope and the coordinated development of regional innovation.

**Data Availability Statement:** Dataset are available from the figshare database (URL: https://doi.org/10.6084/m9.figshare.21900393.v1).

**Funding:** The author(s) received no specific funding for this work.

## Introduction

Endogenous growth theory posits that technological progress is the internal driving force of economic growth and an essential determinant of sustainable economic development [1]. However, due to differences in industrial structures, human capital, and other endowments of innovation resources between economies, the status of innovation and the returns from research and development (R&D) activities differ by country [2]. Some countries are innovation leaders (original innovation), and others are followers (imitation innovation), while economies on the innovation fringe experience prolonged development. Similar to economic development, some studies have pointed out that innovation clubs exist globally [3,4].

Even within a country, the uneven characteristics of innovation development among regions are significant [4]. Particularly in developing countries, internal development differences may increase, even while the country overall maintains a relatively high innovation growth rate. This pattern is mainly because economic or innovation development tends to be concentrated within a certain spatial range, a phenomenon known as "innovation

**Competing interests:** The authors have declared that no competing interests exist.

agglomeration" or "polarization" [5]. Innovation centers formed by innovation polarization, including Bangalore, India, and Shenzhen, China, promote the optimization of regional industrial structures and economic transformation. As one of the determinants of the regional innovation system, the role of the government at all levels in shaping regional patterns cannot be ignored. Some believe moderate government intervention can effectively compensate for market failures and alleviate the negative impact of innovation resource polarization [6]. Others argue that government intervention may destroy the market mechanism and induce rent-seeking behavior, leading to the loss of innovation efficiency [7], further exacerbating regional disparities. The differing perceptions of the government's role suggest that clarifying its influence when exploring innovation development from a regional perspective is vital.

In this study, we supplement the evidence that government intervention affects regional innovation patterns by examining whether the place-based innovation policy promotes regional innovation convergence based on constructing innovative cities in China. On the one hand, while China achieved remarkable innovation development in a few decades, innovation activities have shown significant regional imbalances. Fig 1 shows the dynamic evolution of innovation differences between cities based on the innovation index [8]. As can be seen in the upper portion, the innovation level in China's eastern region is much higher than that in the central and western regions, and this difference tended to increase (from 5.894 to 42.846). The lower portion of Fig 1 shows the change characteristics of the innovation index growth rate, which differs from the index change. In 2004, the innovation growth rate of the eastern cities was 10.2% higher than that of the central cities. By 2020, it was overtaken by 6.1% in central

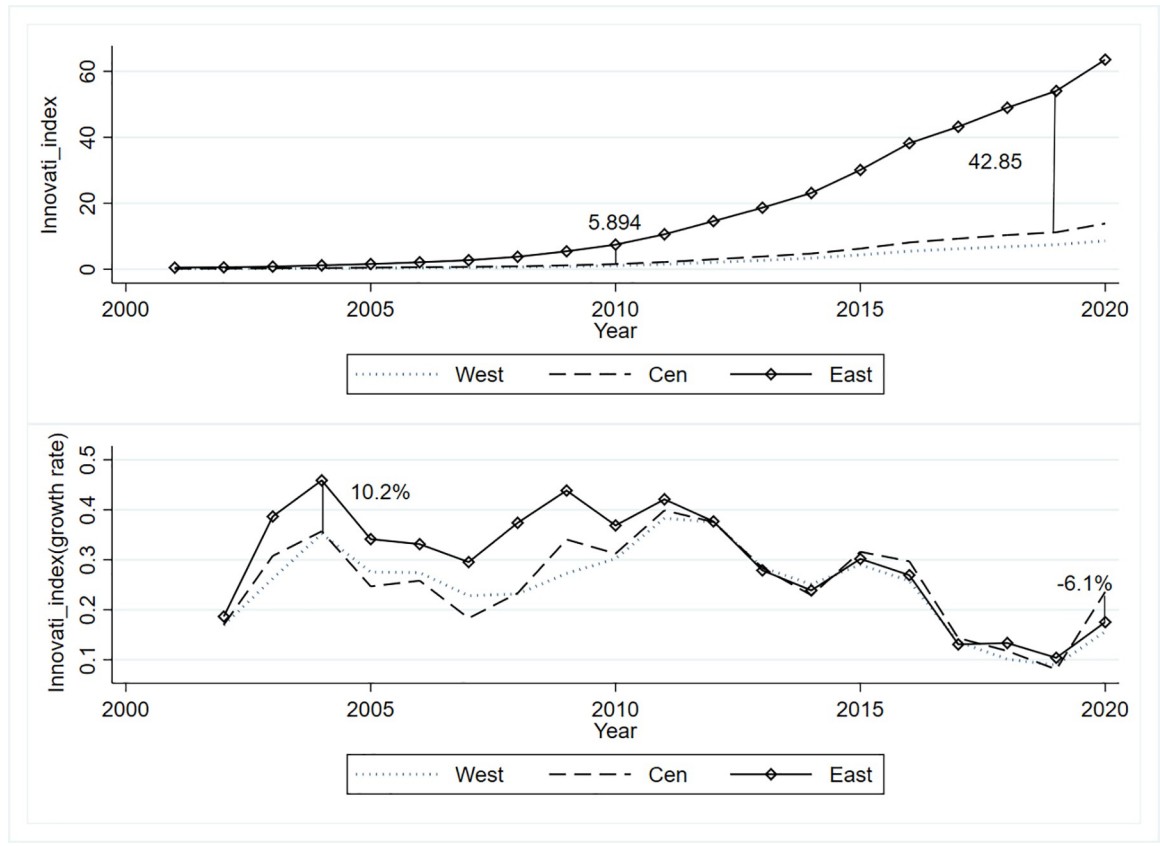

**Fig 1. Dynamic evolution of urban innovation capacity.**

cities. In contrast, the growth rate reflects the possibility of bridging the gap between cities. Therefore, we cannot help but ask whether China's late-developing regions are converging with pioneering areas under the influence of a national innovation policy.

On the other hand, owing to the particularity of China's system, the coordinated interaction between central and local innovation practices has become an important experience and characteristic of China's implementation of innovation-driven development strategies and the construction of an innovative economy [9]. Among them, the innovative city pilot (ICP) is an essential exploratory program for the government and supports urban innovation development; prior studies have confirmed its positive effect on innovation and knowledge flow for pilot cities [10,11]. However, few empirical studies have examined whether such pilot city programs can narrow regional innovation differences. As a typical example of a country with strong government intervention, China not only satisfies the pilot governance premise required by this study but also has the innovative differences between regions that reinforce the need to test policy effects, making China an ideal object for this study.

This study aims to clarify the role of innovative city construction in regional differences, focusing on the core question of whether the innovative city policy promotes regional innovation convergence. Specifically, considering China's innovative city pilot program as a quasi-natural experiment, we evaluate the policy from two aspects: whether it stimulates innovation (basic effect) and accelerates innovation convergence (convergence effect).

Compared with the extant literature, the contributions of this study are as follows. First, the research results reveal the basic effect and convergence effect of innovative city construction on regional innovation, which shows the multiple impacts of the ICP. Second, we find that although the construction of innovative cities promotes innovation convergence among pilot cities, it also slows down the overall regional innovation convergence, revealing the dual character of IPC. Third, following the difference-in-differences (DID) method, we test the moderating effect based on the β convergence model to better reflect the causal relationship between variables. Fourth, the temporal and spatial characteristics of the basic effect and the convergence effect are compared, and the regional heterogeneity of the two effects is captured, presenting the interpretation of dual characters regarding ICP. The results enrich the evaluation research of innovative city construction and provide evidence of government intervention to promote the coordinated development of regional innovation from the perspective of pilot governance. This provides a theoretical basis and practical guidance for the government to summarize the pilot experience further and expand the pilot's scope.

The remainder of this paper proceeds as follows. Section 2 discusses the theoretical background and literature review. Section 3 describes the chosen model and data sources. Section 4 presents the empirical analysis, while Section 5 addresses endogeneity and robustness. In Section 6, we perform a regional heterogeneity analysis. Finally, Section 7 presents the discussion, conclusions, and policy implications.

## Literature review

### Innovation convergence and driving factors

**Innovation convergence.** The spatial convergence and divergence of economies is a long-standing theme of economic growth and development theory [12,13] and an area of debate in regional science and economic geography [4]. The convergence hypothesis refers to when the economic growth rate of latecomer regions grows more quickly than the advanced one, and the per capita income level of latecomer regions converges with that of advanced regions in the long run. Similarly, convergence in the context of innovation refers to innovation production in latecomer regions that have a faster growth rate, and the innovation gap between

regions tends to converge [3,4]. On the contrary, dispersion means that the growth rate of innovation is slower in the latecomer regions, and the gap tends to widen.

Because regional innovation plays a vital role in whether there is convergence in economic growth [4], many scholars have examined the convergence of technology and knowledge among different regions. Asongu and Nwachukwu [14] analyzed the absolute and conditional convergence of science and technology production in 99 countries from 1994 through 2010 and found no absolute β-convergence. They also revealed that developed countries' dominance in scientific knowledge production continued for a longer period. This result is consistent with González et al. [15], who used national scientific production data from 121 developed and developing countries and found no absolute convergence. However, Confraria et al. [16] captured modest features of convergence in scientific productivity between northern and southern countries. Hence, the conclusions are not absolute due to the different research objects and samples. Generally speaking, the convergence characteristics of innovative bodies are more robust at a smaller spatial scale. As Blanco et al. [17] explained, although the innovation model differed among European Union (EU) countries, R&D investment still showed a trend of convergence.

Other scholars examined the convergence of innovation (or knowledge) among regions within a country. For example, Ceh [18] found that the growth rate of patents in the backward states of the United States was faster than that of the traditional core states (Northeast and Midwest regions). O´hUallacha´in and Leslie [19] also identified a spatial convergence of innovation output among U.S. states between 1963 and 1993. There is evidence of innovation convergence recently in China [20–22]. Compared with a general cross-country sample, there is a higher convergence in innovation policies among EU member states or regions within countries. Thus, policies (strategies) formulated by the central or headquarter government may play an important role in the process of regional innovation convergence.

**Driving factors of innovation convergence and regional strategy.** The research on innovation convergence in academia is not limited to the identification of whether convergence exists between different regions. Some literature discusses the role of different factors in the process of regional innovation convergence, taking further measures to narrow the innovation gap between regions.

Hong et al. [23] showed that although regional innovation disparities tend to widen in China; academic-industry cooperation, represented by university-industrial cooperation and industrial-research institute cooperation, helps narrow the regional innovation gap. Using inter-provincial data of China from 2005 to 2016, Yang et al. [20] found that a high-tech industry agglomeration promoted regional innovation convergence, with the knowledge spillover effect as a potential cause of this phenomenon. The study by Tang and Cui [21], based on the city-level data of China, found that the innovation convergence rate of cities within urban agglomerations was significantly higher than that of cities outside urban agglomerations. Yang et al. [22] used panel data from 285 cities in China from 2003 to 2013 and confirmed the acceleration of high-speed rail on regional innovation convergence. In addition, Hou et al. [24] also captured the phenomenon of fiscal science and technology expenditure positively affecting the spatial convergence of regional innovation efficiency.

The current research results reveal the driving factors of regional innovation convergence and provide theoretical support for the government to take targeted measures to promote the coordinated development of regional innovation. China has a relatively high degree of centralization, and the government has taken measures to address uncoordinated regional development. As early as 2000, the central government implemented Western development and promoted coordinated regional development as a strategic task. As a bridge connecting the east and the west, Premier Wen Jiabao first proposed the requirement of the rise of the

midlands. Consequently, the central government included this proposal two years later. Moreover, with increasing demand for innovation-driven economic development, the Chinese government launched innovation pilot projects in cities and regions where conditions permitted, including in prominent cities in the central and western regions.

## Innovative city policy evaluation and its potential impact on innovation differences

Recent research showed that the ICP significantly improved the level of urban innovation. The mechanism of action was not limited to enhancing government fiscal expenditure, industrial agglomeration, and human capital [10], but also included improved knowledge innovation and transformation efficiency from the industry/university/research perspective [11].

In the ICP process, innovative city construction may trigger the existing innovation convergence mechanism. If the pilot area covers the latecomer city, the local government will be encouraged to increase financial investment in science and technology, which will positively affect the spatial convergence of regional innovation efficiency [24]. Including a city in the list of innovative pilot cities will also enhance the region's reputation and attract more highly skilled talent and technology enterprises. As the factors of innovation accumulate to a certain extent, the construction of an innovative city will result in an innovation agglomeration effect. High-tech industrial agglomeration helps enterprises, universities, and other innovation subjects obtain many innovative ideas at low or even zero cost through face-to-face communication [20], as well as deepens industry-university-research collaborative innovation, improving the level of regional innovation. It also realizes inter-regional spillover and transfer of knowledge and technology [23], spreading innovation factors from the more-developed regions to the less-developed regions, thus promoting balanced regional development.

However, the relationship between innovative city construction and regional innovation convergence is not absolutely positive. Some heterogeneity analyses have pointed out that policy effects are more obvious in regions with better economic development [10,11]. The differences could be explained by the following two aspects:

1. Regional Innovation System (RIS). In contrast to the traditional input-output linear innovation model, RIS strengthens the nonlinear path characteristics with a feedback mechanism formed by the interaction of innovation participants [1], where changes in institutions and models are the primary reasons for regional differences [2]. Therefore, even if local governments invest heavily in R&D, they may not produce positive results in the short-term. Carayannis [25] states that the new innovation model focuses on the collaborative interaction among companies, universities, research institutions, governments, and users, which raises the threshold for late-developing cities to benefit from ICP.

2. Innovation absorptive capacity. This knowledge-based perspective emphasizes the importance of external knowledge for innovation [26]. However, not all new external knowledge can be absorbed and utilized, as it often depends on the region's existing accumulation of technology and human capital. Due to the positive externalities of the innovation environment, cities with a higher degree of economic development are more likely to attract the innovative talent needed to absorb new knowledge and develop new technologies [27]. In contrast, less-developed cities have disadvantages in this regard. Therefore, even if the policy is enforced in less-developed regions, the effect of ICP on innovation may be very limited due to poor absorptive capacity.

The above analysis shows that even though the ICP has been piloted in cities of different tiers, the innovation gap between cities is still likely to widen further. Of course, expanding the

pilot cities to inland, non-first-tier cities is itself an attempt to narrow regional differences. Late-developing cities have more room for development [28] and can also accept technology and knowledge transferred from coastal areas [22,29]. Therefore, overall, the effect of ICP on innovation differences is uncertain.

## Policy evolution and research framework

**Policy evolution.** Originating from the practice of Western developed countries in the mid-to-late 1990s, an "innovative city" is the concrete practice of transforming from the traditional investment-driven development mode to the innovation-driven development mode [30]. The concept of the innovative city in China can be traced back to the strategic decision of "building an innovative country" by the China State Council in 2006. Subsequently, many cities put pursued the building of an innovative city. As a pioneer of reform and opening-up, Shenzhen was the first city in China to conduct an innovation pilot (in 2008). In 2009, the National Development and Reform Commission issued the "Notice on Strengthening the Construction of Regional Innovation Basic Capabilities," which recommended improving the basic capabilities of regional innovation by supporting the development of the western region, the revitalization of the old industrial base in the northeast, and the rise of the central region. Hereto, it had set the keynote of "regional coordination" for the subsequent expansion of the pilot scope of an innovative city. In 2010, the number of innovative pilot cities expanded to 42, including 22 eastern cities, 8 central cities, and 12 western cities. By the end of 2018, 78 innovative city construction projects were formed nationwide, covering 31 provinces, municipalities, and autonomous regions in mainland China. The list of cities in the innovative city pilot program is shown in Table 1.

Currently, to become a pilot city, a city must go through the proper procedure of local recommendation (provincial government), material review, and expert review. Finally, the Ministry of Science and Technology publishes the list of pilot cities, and the project acceptance begins after several years (2–3 years). During this period, pilot cities can explore and formulate development plans according to local conditions. Some provinces in the eastern region have an absolute advantage with a higher number of pilot cities: Zhejiang Province has six approved cities (prefecture-level cities). In comparison, some western provinces have only one to two pilot cities. Regarding development quality, the "National Innovative City Innovation Capability Monitoring Report 2020" by the China Institute of Science and Technology Information shows that among the top 30 innovation capability index rankings for cities (including four municipalities), 20 cities are located in the east. Hence, significant differences in urban innovation development between regions remain.

**Table 1. Innovative city pilot list.**

| Year | Eastern China | Central China | Western China |
|---|---|---|---|
| 2008 | Shenzhen | | |
| 2010 | Beijing (Haidian District), Tianjin (Binhai New District), Shanghai (YangpuDistrict), Dalian, Qingdao, Xiamen, Shenyang, Guangzhou, Nanjing, Hangzhou, Jinan, Suzhou, Wuxi, Yantai, Tangshan, Ningbo, Jiaxing, Shijiazhuang, Changzhou, Fuzhou, Haikou | Changsha, Harbin, Hefei, Taiyuan, Jingdezhen, Wuhan, Nanchang, Luoyang | Chongqing (Shapingba District), Xi'an, Guiyang, Kunming, Chengdu, Baotou, Shihezi, Lanzhou, Nanning, Baoji, Changji, Yinchuan |
| 2011 | Lianyungang, Qinhuangdao, Zhenjiang | Changchun | Xining, Hohhot |
| 2012 | Nantong | Zhengzhou | Urumqi |
| 2013 | Yangzhou, Taizhou, Yancheng, Jining, Huzhou | Yichang, Xiangyang, Pingxiang, Nanyang | Zunyi |
| 2018 | Xuzhou, Quanzhou, Weifang, Longyan, Jinhua, Foshan, Shaoxing, Dongg, Dongying | Zhuzhou, Hengyang, Jilin, Ma'anshan, Wuhu | Yuxi, Lhasa, Hanzhong |

**Convergence type and research framework.** The policy's convergence effect can be viewed from different perspectives. First, we examined whether pilot cities have a higher rate of innovation convergence than non-pilot cities (Convergence_1). Second, we considered the spatial spillover effects (basic and convergence effects) of pilot policy because the spillover distance affects the number of cities within the radius of the innovation center (Convergence_2). Additionally, the radii of influence of basic and convergence effects may be different. Some cities experience an increase in the level of innovation due to the close location of an innovative city (innovation center); however, because the spillover effect is too small, it may not be enough to accelerate the convergence to developed regions. Finally, we investigated potential regional heterogeneity in policy effects. Innovation differences between Chinese cities appear not only in regions but also in a trend toward further expansion of innovation differences in cities within regions. Therefore, we also examined whether the pilot policy's basic and convergence effects is significant in different regions (Convergence_3).

## Methods and data

### Empirical model

We used the β-convergence method of Baumol [12] and Sala-I-Martin [13] to test the convergence of urban innovation in China. Referring to Sonn and Park [31] and Yang et al. [22], we constructed the following model to examine China's absolute β-convergence of urban innovation:

$$D.ln\_Y_{it} = \alpha_i + \mu_t + \beta_0 L.ln\_Y_{it} + \varepsilon_{it}, \tag{1}$$

where $i$ and $t$ represent the city and year, respectively. $L.ln\_Y_{it}$ is the lag term of the urban innovation index and $D.ln\_Y_{it}$ is the first-order difference term of the innovation index. $\alpha_i$ and $\mu_t$ represent the individual features that do not change with time, and the time features that do not change with individuals, respectively. $\varepsilon_{it}$ is the random disturbance term. Whether there is innovation convergence between cities depends on the coefficients $\beta_0$, where only significantly negative values show signs of convergence. Considering that each region has unique basic conditions of economic development and innovation, we use the conditional β-convergence:

$$D.ln\_Y_{it} = \alpha_i + \mu_t + \beta_0 L.ln\_Y_{it} + \gamma Z'_{it} + \varepsilon_{it}. \tag{2}$$

Eq (2) added the following control variables, $Z'_{it}$, and may affect the level of urban innovation to Eq (1), including the industrial structure, natural growth rate, technological progress rate, government fiscal spending on science and technology, traffic conditions, development of enterprises and financial institutions, the level of human capital, communication, and opening to the outside world.

We use this as a quasi-natural experiment, dividing pilot cities into an experimental group and the other cities as a control group to examine the impact of ICP on innovation convergence. To account for the differences in time since the pilot cities were established, we constructed a time-varying DID model:

$$D.ln\_Y_{it} = \nu_i + \mu_t + L.ln\_Y_{it} + \beta_1 Policy_{it} + \beta_2 Policy_{it} \times L.ln\_Y_{it} + \gamma Z'_{it} + \varepsilon_{it}. \tag{3}$$

Eq (3) added the policy effect (*Policy, Treatment×Time*) and the interaction term of *Policy*$_{it}$×*L.ln_Y*$_{it}$ based on Eq (1): if city $i$ belongs to the treatment group, then the treatment value is 1 and 0 otherwise. Time is a dummy variable for the time before and after the policy implementation, with 0 before the policy is implemented and 1 after implementation.

## Variables and data sources

We used panel data from 275 cities in China from 2003 to 2020. Most existing studies adopt the number of patents granted or the number of patents granted per capita to measure innovation [10,22]. However, such indicators are homogeneous because they cannot measure the social value of different patents. In addition, China's patent innovation "bubble" is severe. Therefore, we used the innovation index in the "Report on Innovation Capability of China's Cities and Industries" by Kou [8]. The index uses updated information on the legal status of the micro-invention patents granted by the State Intellectual Property Office of China. The patent value is calculated using the patent update model, which has strong objectivity and authority. It should be pointed out that the innovation index was only updated in 2016.

The index after 2017 is supplemented proportionally (the ratio of the current year to the previous year), according to the number of invention patents granted, which comes from Chinese Research Data Services. Industrial structure, natural growth rate, R&D investment, financial level, human capital, enterprise development, communication, opening-up, and transportation are obtained from the China Urban Statistical Yearbook. In addition, the inter-city distances required for subsequent analysis were the spherical distances between points, calculated using ArcGIS. The average urban slope (*slope*) and average urban elevation [22] (*elevation*) were processed using ArcGIS based on SRTM data (DEM spatial distribution data of altitudes in China) downloaded from the Chinese Academy of Sciences website. A description of the variables is shown in Table 2.

## Estimation results

### Benchmarking

In Table 3, Column 1 shows the regression result of absolute convergence, where the coefficient of the lag term $L.ln\_Y$ is -0.1186, indicating absolute convergence in China's urban innovation. Column 2 shows the regression result after adding the control variables. The coefficient of $L.ln\_Y$ is -0.1516, which indicates conditional convergence in urban innovation. After controlling for the factors that potentially affect innovation, the absolute value of the coefficient increases (the speed of convergence accelerates), which means that the growth rate of innovation is not only negatively correlated with the initial innovation level, but also affected by industrial structure, R&D input, and other factors. In Columns 3 and 4, after *Policy* are added, the coefficient of $L.ln\_Y$ increases slightly, indicating that ICP may intensify regional innovation divergence. In Columns 5 and 6, we use the logarithm of the innovation index as the explained variable. The effect sizes of the policy are all positive (0.3055 / 0.1500), demonstrating that the ICP positively affects urban innovation and verifying the basic effects of the policy, which is consistent with the finding of Zhou and Li [10].

### Moderating and mediating effects

The coefficient of $L.ln\_Y \times Policy$ in Column 7 of Table 4 is significantly negative (-0.0181), indicating faster innovation convergence in cities conducting pilots. This result also holds in Column 8 (-0.0192) after adding the control variables. Fig 2 more intuitively shows the positive impact of the pilot policy on the convergence rate of urban innovation; the shaded area represents the 95% confidence interval. Combined with the results of Columns 3 and 4 in Table 1, it can be concluded that ICP promotes innovation convergence among pilot cities but intensifies the innovation divergence of the whole sample (convergence_1 has been tested). This may result from a widening innovation gap between the cities undergoing ICP and those that have yet to do so. Columns 9–11 aim to test the policy impact mechanism. In Columns 10 and 11, the policy coefficients (0.1500/0.1544) after adding R&D funds and R&D personnel are significantly

**Table 2. Descriptive statistics.**

| Variable | Variable definitions | Mean | Std. Dev. | Min | Max |
|---|---|---|---|---|---|
| **Explained variables** | | | | | |
| *innovation index* | Measured by Kou [8] | 0.0461 | 1.9376 | -5.2729 | 7.0575 |
| *Patent_invention* | the number of invention patents granted [9] | 3.8587 | 2.2757 | -6.9077 | 10.3421 |
| *Patent _total* | the number of all patents granted [9] | 6.4694 | 1.8426 | 0.6931 | 12.3100 |
| **Core explanatory variable** | | | | | |
| *Policy* | dummy variable, with ICP is 1, otherwise is 0 | 0.1158 | 0.3201 | 0 | 1 |
| **Control variables** | | | | | |
| *Industry_sec* | the proportion of secondary industries [21] | 0.4729 | 0.1124 | 0.0276 | 0.9097 |
| $n + g + \delta$ | the sum of the natural growth rate (n), technological progress rate (g), and depreciation rate ($\delta$), where $g+\delta$ is equal to 5% [22] | 2.2815 | 0.5164 | -2.9967 | 3.8248 |
| *R&D_exp* | government fiscal spending on science and technology [21] (10,000 yuan) | 9.3701 | 1.8441 | 3.8712 | 15.5293 |
| *H_cap* | the number of college students per 10,000 people [20,21] | 4.4753 | 1.1193 | -0.5242 | 7.1787 |
| *Finan* | the sum of deposits and loans from financial institutions (10,000 yuan) | 16.7776 | 1.2377 | 13.3523 | 21.1829 |
| *C_profit* | the total profit of industrial enterprises above a specific size (annual main business income is more than 20 million yuan; (10,000 yuan)) | 5.5071 | 0.9989 | 1.9839 | 8.4233 |
| *Commu* | the total number of mobile phones and Internet users [21] (10,000 households) | 13.3299 | 1.5551 | 6.5696 | 17.7593 |
| *Open* | the actual use of foreign capital (USD 10 000) | 9.3026 | 2.6079 | -6.9077 | 14.1523 |
| *Trans* | the total passenger of transportation (including roads, waterways, and flights; 10,000 people) [22] | 8.4480 | 0.9965 | 0.4055 | 12.5668 |
| **Instrumental variable** | | | | | |
| *Slope* | the average urban slope | 7.9787 | 1.2657 | 4.3791 | 10.0807 |
| *Elevation* | the average urban elevation [22](m) | 13.0512 | 1.4005 | 9.5953 | 15.6591 |
| **Other variable** | | | | | |
| *R&D_talent* | The number of scientific research and technical service employees | 8.2991 | 1.0789 | 4.6051 | 13.6048 |

Note: Except for Industry_sec and Policy, all other variables take the logarithm (If the value is 0, 0.001 is used instead).

**Table 3. Benchmark regression results.**

| VARIABLES | (1) | (2) | (3) | (4) | (5) | (6) |
|---|---|---|---|---|---|---|
| | D.ln_Y | | | | ln_Y | |
| *L.ln_Y* | -0.1186*** (0.0112) | -0.1516*** (0.0104) | -0.1178*** (0.0134) | -0.1499*** (0.0104) | | |
| *Policy* | | | -0.0119 (0.0125) | -0.0352*** (0.0127) | 0.3055*** (0.0784) | 0.1500** (0.0717) |
| *Constant* | 0.0187 (0.0230) | -2.6960*** (0.5263) | 0.0203 (0.0231) | -2.6989*** (0.5255) | -1.9081*** (0.0362) | -7.9423*** (1.7119) |
| *Control variable* | no | yes | no | yes | no | yes |
| *Time fixed effect* | yes | yes | yes | yes | yes | yes |
| *Individual fixed effect* | yes | yes | yes | yes | yes | yes |
| *R-squared* | 0.287 | 0.325 | 0.287 | 0.326 | 0.888 | 0.908 |
| *Observations* | 4,668 | 4,457 | 4,668 | 4,457 | 4,944 | 4,732 |

Note: Robust standard errors in parentheses

***p<0.01

**p<0.05

*p<0.1.

**Table 4. Test of moderating and mediating effects.**

|  | (7) | (8) | (9) | (10) | (11) |
|---|---|---|---|---|---|
| VARIABLES | D.ln_Y | D.ln_Y | ln_Y | ln_Y | ln_Y |
| L.ln_Y | -0.1160*** | -0.1482*** |  |  |  |
|  | (0.0114) | (0.0104) |  |  |  |
| Policy | 0.0353** | 0.0142 | 0.2332*** | 0.1500** | 0.1544** |
|  | (0.0178) | (0.0184) | (0.0742) | (0.0717) | (0.0705) |
| L.ln_Y× Policy | -0.0181*** | -0.0192*** |  |  |  |
|  | (0.0057) | (0.0060) |  |  |  |
| ln(R&D_exp) |  | 0.0582*** |  | 0.2233*** |  |
|  |  | (0.0086) |  | (0.0362) |  |
| ln(R&D_talent) |  |  |  |  | 0.0942*** |
|  |  |  |  |  | (0.0324) |
| Constant | 0.0237 | -2.6169*** | -9.9347 | -7.9429*** | -8.5033*** |
|  | (0.0232) | (0.5252) | (1.9320) | (1.7119) | (1.6883) |
| Control variables | no | yes _ | yes | yes | yes _ |
| Time fixed effect | yes | yes | yes | yes | yes _ |
| Individual fixed effect | yes | yes | yes | yes | yes _ |
| R-squared | 0.288 | 0.327 | 0.899 | 0.908 | 0.907 |
| Observations | 4,668 | 4,457 | 4,732 | 4,732 | 4,732 |

Robust standard errors in parentheses

***p<0.01

**p<0.05

*p<0.1.

below the baseline model coefficient (0.2332). These results support the idea that the ICP can influence urban innovation through two basic intermediary mechanisms: increasing government technical expenditures and the number of scientific and technological personnel.

## Dynamic effect test

The regression results of benchmark testing and the moderating effect reflect the average impact of the ICP's basic and convergence effects. However, they do not reflect the difference in the impact of the policy in different periods. Furthermore, the parallel trend assumption for the treatment and control groups should be satisfied when using the DID method. Therefore, we examined the dynamic effects of the ICP and constructed the following model:

$$D.ln\_Y_{it} = v_i + \mu_t + L.ln\_Y_{it} + \delta_k \sum\nolimits_{k\geq-4}^{+12} Treatment_{it}^k * L.ln\_Y_{it} * year_{it}^{2008+k} + \gamma Z'_{it} + \varepsilon_{it}. \quad (4)$$

$$ln\_Y_{it} = v_i + \mu_t + L.ln\_Y_{it} + \delta_k \sum\nolimits_{k\geq-4}^{+12} Treatment_{it}^k * year_{it}^{2008+k} + \gamma Z'_{it} + \varepsilon_{it}. \quad (5)$$

*Year* is a dummy variable equal to 1 in the policy pilot period and 0 otherwise. The other variables are consistent with those in the baseline model. We note that the base year is before the policy implementation in Shenzhen (2007). We illustrate the trend in the first three years (removing the base period) and 12 years after the policy implementation in Fig 3. The abscissa is the relative time of policy implementation, and the ordinate is the estimated coefficient of *Treatment×L.ln_Y×year* and *Treatment×year*. The left portion of Fig 3 represents the policy convergence effect change during different periods. From the figure, the coefficients before the policy implementation are not significant, ensuring that the common trend assumption of the treatment and control groups is satisfied. The coefficient is significantly negative from the fifth

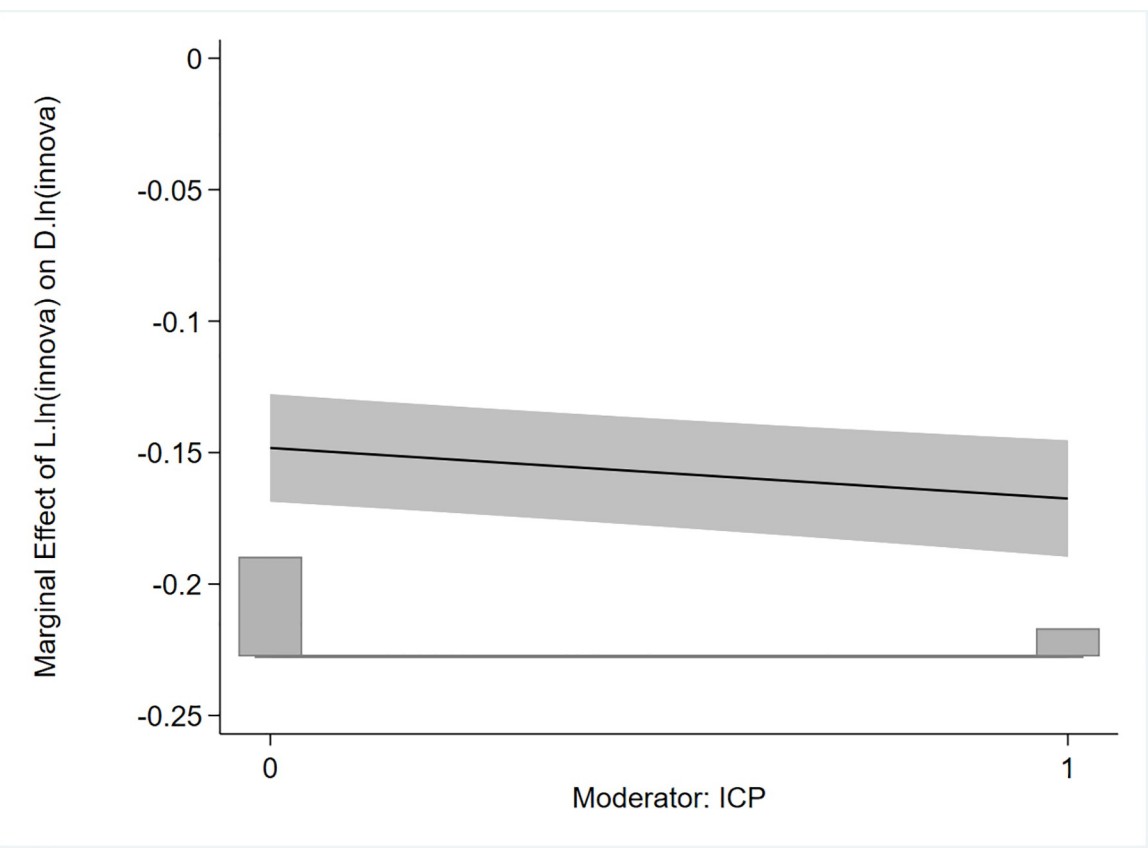

**Fig 2. Moderating effect of the pilot policy on innovation convergence.**

year ($L.ln\_Y \times Policy$) after the policy pilot, indicating that the ICP has a lag period in promoting innovation convergence in pilot cities. The development of innovation facilitated by policy pilots may not accelerate the convergence of innovation development to a steady state in the short-term. Thus, we constructed Eq (5) to examine the time difference between the basic policy effect and the convergence effect.

As shown in Fig 3, the basic effect of the ICP has a positive impact for two years following the policy's implementation. In general, the above comparison demonstrates that the convergence effect of the ICP on innovation development lags slightly behind the basic effect. Considering the small number of pilot cities in the early stage (in 2008, only one city—Shenzhen, was an ICP), it takes a certain amount of time from the start of the pilot (input) to see an increase in innovation output. It is not surprising that both the basic effect and the convergence effect have lag periods.

## Spatial spillover effect test

The closer the distance between ordinary cities and innovation centers, the more significant the spillover effect of innovation [9,28]. We took pilot innovative cities as regional innovation centers to test the spillover effects of the pilot policy using two approaches.

Method (1): As in Yang et al. [22], we first set the spatial distance (spherical distance) interval (0–120 km) and then added 60 km at a time. Second, after calculating the distance between ordinary cities and the nearest innovation centers, we included the number of innovation centers in the study. Specifically, following the benchmark model, if there is only one innovation

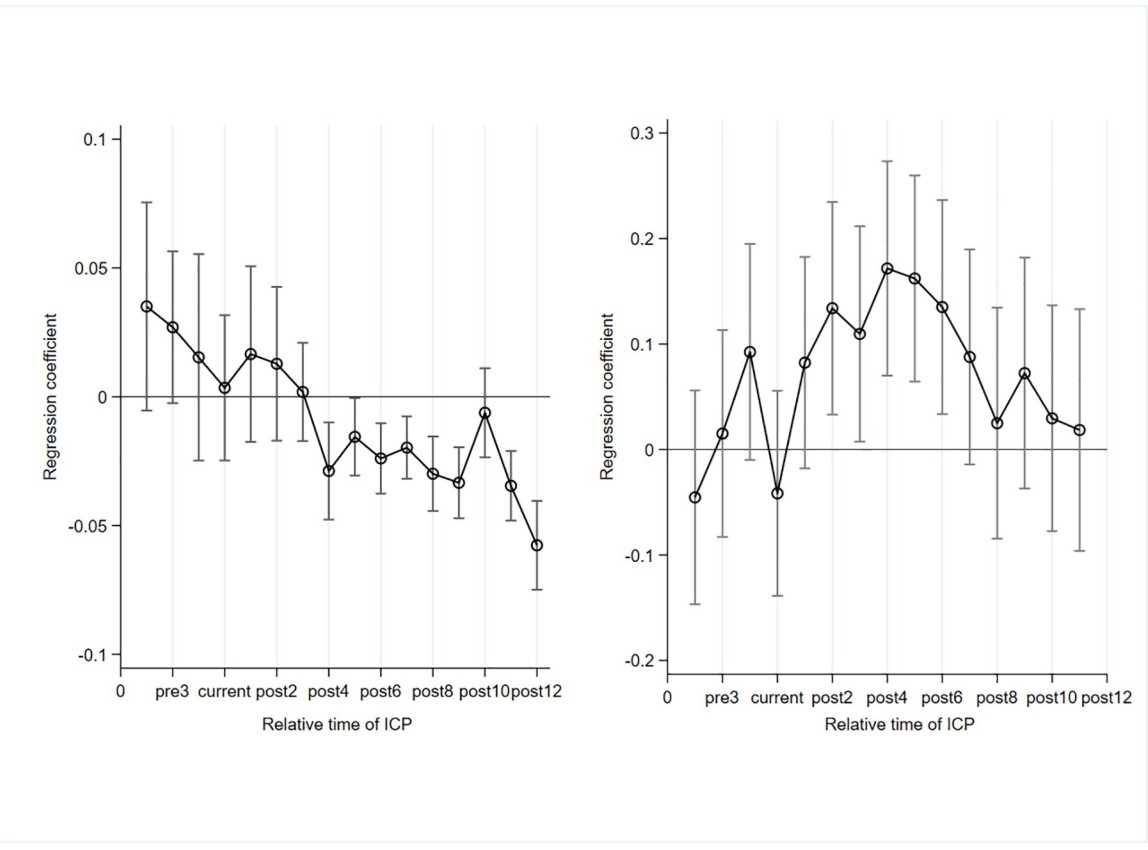

**Fig 3. Dynamic evolution of the convergence effect and basic effect.**

city within the distance interval, then *Policy_spillas* equals 1, and 0 otherwise. If there are two innovation cities in the same distance interval, then we use the policy implementation date for the first city as the start time of the spillover effect, where the *Policy_spill* value of the following time is 2 and 0 otherwise. By analogy, if there are three or more innovative cities in the distance interval, the value *Policy_spill as* equals 3.

We conducted the regression by deleting the sample of cities in which the ICP is implemented; Table 5 reports the results. In Columns 12–14, within 0–120 km, the basic and convergence effects of the ICP are significant, and the signs are consistent with the previous ones. In Column 14, which used the logarithm of the innovation index as the dependent variable, the policy coefficient (0.2485) is higher than that in Column 6 (0.1500). Hence, the policy spillover effect on the innovation growth rate of surrounding cities is stronger than the improvement of pilot cities on their own. In contrast, Columns 15–17 show the regression results in the interval of 120–180 km. Although the lag term ($L.ln\_Y$) is still significantly negative, the coefficients of policy and its interaction term ($L.ln\_Y \times Policy\_spill$) are not significant. The results indicate that the spatial spillover distance of the innovation center is 120 km, regardless of whether it is the basic effect or the convergence effect.

Method (2): To verify the robustness of the results of Method (1), we construct the following model:

$$D.ln\_Y_{it} = v_i + \mu_t + \beta_0 Policy_{it} + \sum\nolimits_{s=180}^{360} \delta_s N_{it}^s I + \delta_{s0} N_{it}^{s0} I + \gamma Z'_{it} + \varepsilon_{it} \qquad (6)$$

**Table 5. Spatial spillover effect of the pilot policy (Plan 1).**

| | (12) | (13) | (14) | (15) | (16) | (17) |
|---|---|---|---|---|---|---|
| Distance (km) | 0–120 | | | 120–180 | | |
| VARIABLES | D.ln_Y | D.ln_Y | ln_Y | D.ln_Y | D.ln_Y | ln_Y |
| $L.\ln\_Y$ | -0.1230*** | -0.1167*** | | -0.1177*** | -0.1166*** | |
| | (0.0124) | (0.0135) | | (0.0122) | (0.0121) | |
| Policy_spill | 0.0341** | 0.0456*** | 0.2485*** | -0.0064 | 0.0047 | -0.0795 |
| | (0.0140) | (0.0146) | (0.0502) | (0.0140) | (0.0132) | (0.0522) |
| $L.\ln\_Y \times Policy\_spill$ | | -0.0125** | | | -0.0086 | |
| | | (0.0060) | | | (0.0079) | |
| Constant | -0.0713** | -0.0570** | -8.3389*** | -0.0592*** | -0.0568* | -8.0946*** |
| | (0.0304) | (0.0330) | (1.7206) | (0.0299) | (0.0299) | (1.7884) |
| Control variables | yes | yes | yes | yes | yes | yes |
| Time fixed effect | yes | yes | yes | yes | yes | yes |
| Individual fixed effect | yes | yes | yes | yes | yes | yes |
| R-squared | 0.283 | 0.284 | 0.902 | 0.281 | 0.282 | 0.898 |
| Observations | 3,746 | 3,746 | 3,772 | 3,746 | 3,746 | 3,772 |

Note: Robust standard errors in parentheses

***p<0.01

**p<0.05

*p<0.1.

Eq (6) introduces new control variables, $N_{it}^{s0}$ and $N_{it}^{s}$ to Eq (1), where $s$ represents the distance between cities (km, $s \geq 180$). Specifically, if there is an innovation pilot city within the spatial range of city $i$ (0, s] in year $t$, then $N_{it}^{s} = 1$; otherwise $N_{it}^{s} = 0$. For example, $N_{it}^{180}$ indicates whether there is an innovation city within a spatial range of 0–180 km from city $i$ in year $t$. $s0$ is the initial distance dummy variable; if there is an innovation pilot city $N_{it}^{s0}$ within 0–120 km, then $N_{it}^{s0} = 1$ and 0 otherwise. For the convergence spillover effect, we add the interaction terms $L.\ln\_Y_{it} \times \Sigma_{s=120}^{360} \delta_s N_{it}^{s}$ and $L.\ln\_Y_{it} \times N_{it}^{s0}$ to Eq (6), where the other variables are the same as in Method (1). For different distance intervals, we performed the regression in batches ($I$ is an indicative function; when the regression belongs to the distance interval batch, the value is 1 and 0 otherwise). We used $D.\ln\_Y_{it}$ and $L.\ln\_Y_{it}$ as the explained variables to perform the regression, as shown in Table 6. We tested the spatial spillovers of ICP by comparing the significance of the $\delta_s$ under different thresholds.

The results in Table 6 are consistent with those of Columns 12, 13, 15, and 16 from Method (1). The coefficient of $N_{it}^{s} \times L.\ln\_Y$ in Column 18 is negative (-0.0109), although it is only significant at the 10% level. However, combined with Method (1), the policy spillover effect on innovation convergence in surrounding cities is insignificant within 0–120 km. Consistent with the conclusions in Column 16, the results in Column 19 further verify that within the spatial range of 120–180 km, the convergence effect of the policy on the innovation development of surrounding cities is not significant. Specifically, the basic effect of the policy gradually decreases as the distance to the innovative city increases. In contrast to the results in Columns 14 and 17, the basic effect of national innovation cities on surrounding cities could extend to 180 km. In general, the results in Tables 5 and 6 show that ICP has a significant innovation spillover effect, which not only improves the innovation level of surrounding cities but also promotes innovation convergence between these cities and pilot cities. The phenomenon of innovation agglomeration can promote knowledge spillover and technology transfer to achieve further convergence of innovation between cities. However, it is also important to

**Table 6. Convergence effect of the pilot policy in spatial spillover (Plan 2).**

| VARIABLES | (18) D. ln_Y | (19) D.ln_Y | (20) ln_Y | (21) ln_Y | (22) ln_Y |
|---|---|---|---|---|---|
| $N^s_{it}$(0–120 km) | 0.0512*** (0.0183) | | 0.2144*** (0.0785) | | |
| $N^s_{it}$(120–180 km) | | -0.0279** (0.0135) | | 0.1064* (0.0601) | |
| $N^s_{it}$(180–240 km) | | | | | 0.0506 (0.0533) |
| $N^s_{it} \times$ L.ln_Y (0–120 km) | -0.0109* (0.0059) | | | | |
| $N^s_{it} \times$ L.ln_Y (120–180 km) | | -0.0092 (0.0056) | | | |
| Constant | -2.4563*** (0.5205) | -2.4308*** (0.5198) | -8.0877*** (1.6838) | -7.9326 (1.6886) | -7.9406 (1.7105) |
| Observations | 4,440 | 4,440 | 4,714 | 4,714 | 4,714 |
| R-squared | 0.349 | 0.348 | 0.909 | 0.908 | 0.908 |

Note: Robust standard errors in parentheses

***p<0.01

**p<0.05

*p<0.1.

point out that the spillover effect is only effective within a certain spatial range. Hereto the convergence_2 has been tested.

## Endogeneity and robustness tests

### Two-stage least squares method

Although our results show that the policy can improve the speed of urban innovation convergence, the selection of innovative pilot cities often prioritizes cities with superior economic foundations and agglomeration of innovative elements. Therefore, there is a two-way causal relationship between policy implementation and urban innovation. Thus, we added the interaction term between geographical feature [22] (the average urban slope, (*slope*); the average urban elevation, (*elevation*)) and the year dummy variable as an instrumental variable for the pilot policy. Although geographic indicators, such as slope, influence construction and traffic commuting within a city, this effect gradually diminishes with technology development [32]. In the short-term, geographic variables generally do not change over time and can be understood as exogenous. However, the central government often prefers areas with good infrastructure (the average slope affects land and engineering construction) for the pilot construction of innovation cities; that is, there may be a "geographical prejudice" in selecting a pilot city.

We also verified the rationality of our choice of instrumental variables through a strict measurement inspection. First, we performed a regression with $ln\_Y_{it}$ as the explained variable and geographic indicators as the explanatory variable. The results in Table 7 indicate no significant association between the two. Columns 26 and 28 show the first-stage regression results. The coefficients of *Slope × Year* and *Elevation × Year* are negative, indicating an inverse relationship between a city's average slope (or elevation) and whether it is an innovative city. It also reflects that geographical features within cities may influence whether a city is included on the pilot list. The under-identification test (Kleibergen-Paap rk Wald F is 10.221 and 43.467,

**Table 7. Endogeneity test.**

| VARIABLES | (23) | (24) | (25) | (26) | (27) | (28) | (29) | (30) |
|---|---|---|---|---|---|---|---|---|
| | | | 2 SLS | | | | PSM-DID | |
| | | | Second-stage | First-stage | Second-stage | First-stage | | |
| | ln_Y | ln_Y | D.ln_Y | Policy | D.ln_Y | Policy | D.ln_Y | D.ln_Y |
| L.ln_Y | | | -0.0921*** (0.0251) | 0.0511*** (0.0071) | -0.1111*** (0.0186) | 0.2932*** (0.0377) | -0.1168*** (0.0172) | -0.1360*** (0.0186) |
| Policy | | | -0.8955*** (0.4508) | | 0.5843 (0.4968) | | 0.0411 (0.0230) | -0.0275 (0.0162) |
| L.ln_Y ×Policy | | | | | -0.2499* (0.1395) | | -0.0313** (0.0154) | -0.0296** (0.0146) |
| Slope ×Year | 0.7622 (2.1228) | | | -0.9608*** (0.2914) | | -1.4697** (0.7010) | | |
| Slope ×Year ×L.ln_Y | | | | | | -0.0194*** (0.0049) | | |
| Elevation ×Year | | 0.0019 (0.0061) | | | | -0.0056** (0.0024) | | |
| Elevation ×Year× L.ln_Y | | | | | | 0.0046** (0.0019) | | |
| Kleibergen-Paap rk LM | | | 10.221*** | | 43.467*** | | | |
| Kleibergen-Paap rk Wald F | | | 10.869 (8.96) | | 10.304 (7.56) | | | |
| Control variables | yes | yes | yes | yes | yes | yes | no | yes |
| Time fixed effect | yes | yes | yes | yes | yes | yes | yes | yes |
| Individual fixed effect | yes | yes | yes | yes | yes | yes | yes | yes |
| Observations | 4,732 | 4,732 | 4457 | 4457 | 4,457 | 4,457 | 900 | 900 |

Note: Robust standard errors in parentheses

***p<0.01

**p<0.05

*p<0.1.

rejecting the null hypothesis) and weak instrumental variable test (all statistical values above the 15% maximal IV size) results also show that the selected instrumental variable does not indicate problems with insufficient and weak instrumental variables. From the second-stage regression, the coefficients of the lag term $L.ln\_Y$ are all significantly negative, and the pilot policy coefficient in Column 25 is -0.8955 (significant at the 1% level), which is also consistent with Eq (4) results. The coefficients of $L.ln\_Y×Policy$ in Column 27 are significantly negative, indicating that the underlying two-way causality does not significantly affect the innovation convergence effect of the policy.

## PSM- DID

To further overcome the influence of sample selection bias on the estimation results, we applied the propensity score matching (PSM) method to match the samples. All the control variables in Eq (2) are selected as a covariate.

We used a year-by-year method to perform nearest neighbor (neighbor (5)) matching. The standardized deviation values (% bias) of each control variable in the treatment group and the control group in each year were almost all less than 20% [33]. Finally, we combined the city-level data after matching each for the regression. In Columns 29 and 30, the coefficient of the interaction term is negative (-0.0313/-0.0296), regardless of whether we added the control

variables, and *L.ln_Y×Policy* is significant at the 5% level, which further shows that the original conclusions are robust.

## Alternative policy interference and replacing the dependent variables

Avoiding the interference of other policies or shocks is an important premise of using a DID model to ensure a robust analysis. The period for the implementation of the ICP is also a period when other relevant policies (which may affect innovation) are promulgated or implemented. For example, with the maturity of new-generation information technologies, such as the Internet of Things and cloud computing, and the need to build a modern city, the Chinese government proposed the smart city concept in 2009 and officially established smart city pilots in 2012. Second, since Beijing Zhongguancun became the first national independent innovation demonstration zone in 2009, the Chinese government successively approved more than ten national independent innovation demonstration zones, most of which consist of several representative cities. To test the degree of interference of such policies, we added the dummy variables *Policy_S* for the pilot smart cities and *Policy_N* for the national independent innovation demonstration zones to Eq (4). Columns 28 and 29 in Table 8 present the results. The coefficient of *L.ln_Y×Policy* is still significantly negative at the 5% level (-0.0165 / -0.0157), which means that the innovation convergence of ICP identified above has not been inhibited by other regional policies.

In addition, we replaced the dependent variable with the total number of patents (inventions, utility models, and designs) and the number of invention patents as explained variables

**Table 8. Robustness test.**

| VARIABLES | (31) | (32) | (33) | (34) | (35) | (36) |
|---|---|---|---|---|---|---|
| | Policy interference | | Dependent variable_replacement _ | | | |
| | Smart_city | NIDZ | | | | |
| | D.ln_Y | D.ln_Y | Patent _ total | | Patent_invention | |
| L.ln_Y | -0.1314*** (0.0088) | -0.1311*** (0.0088) | -0.4042 *** (0.0181) | -0.4042*** (0.0180) | -0.2692*** (0.0121) | -0.2687*** (0.0121) |
| Policy | 0.0081 (0.0096) | 0.0078 (0.0198) | -0.0607 *** (0.0290) | 0.1904*** (0.0992) | -0.0665*** (0.0157) | 0.2762*** (0.0873) |
| L.ln_Y × Policy | -0.0165** (0.0076) | -0.0157** (0.0079) | | -0.0383** (0.0156) | | -0.0401*** (0.0102) |
| Policy_S | 0.0182 (0.0146) | | | | | |
| L.ln_Y × Policy_S | -0.0068 (0.0059) | | | | | |
| Policy_ N | | 0.0148 (0.0146) | | | | |
| L.ln_Y × Policy_N | | -0.0071 (0.0062) | | | | |
| Control variables | yes | yes | yes | yes | yes | yes |
| Time fixed effect | yes | yes | yes | yes | yes | yes |
| Individual fixed effect | yes | yes | yes | yes | yes | yes |
| Observations | 4,457 | 4,457 | 4,396 | 4,396 | 4,454 | 4,454 |
| R-squared | 0.348 | 0.348 | 0.335 | 0.336 | 0.344 | 0.346 |

Note: Robust standard errors in parentheses

***p<0.01

**p<0.05

*p<0.1.

for the regression. The significance and direction of the coefficients of *L.ln_Y×Policy* remain unchanged, as shown in Columns 33–36. These results show that the convergence effect of the pilot policy for innovative cities is still robust after excluding the impact of relevant policies and replacement indicators.

## Placebo test

Although we checked for some policy shocks that could affect the estimates, other unobserved shocks may affect the estimates. Therefore, we randomized the pilot sample and pilot time. Specifically, in scheme (1), keeping the pilot cities unchanged, we randomly selected a time (year) sample from the variable year (2003–2020) as the implementation time and generated a false-policy variable on this basis. Scheme (2) draws on Cai's study [34], where we randomly selected cities from 275 cities, divided them into innovative pilot cities, and constructed false variables based on this. If no other shocks affect the original estimates, then the results of the randomization process should show that the false-policy dummy variables we constructed do not affect $D.ln\_Y_{it}$. Fig 4 shows the coefficient kernel densities and corresponding p-value distributions for the 500 false treatment groups. For scheme (1) (left in Fig 4) or scheme (2) (right in Fig 4), the mean value of the randomly generated interaction term coefficients is near 0, and most of the P values are greater than 0.1, further indicating the robustness of our conclusions.

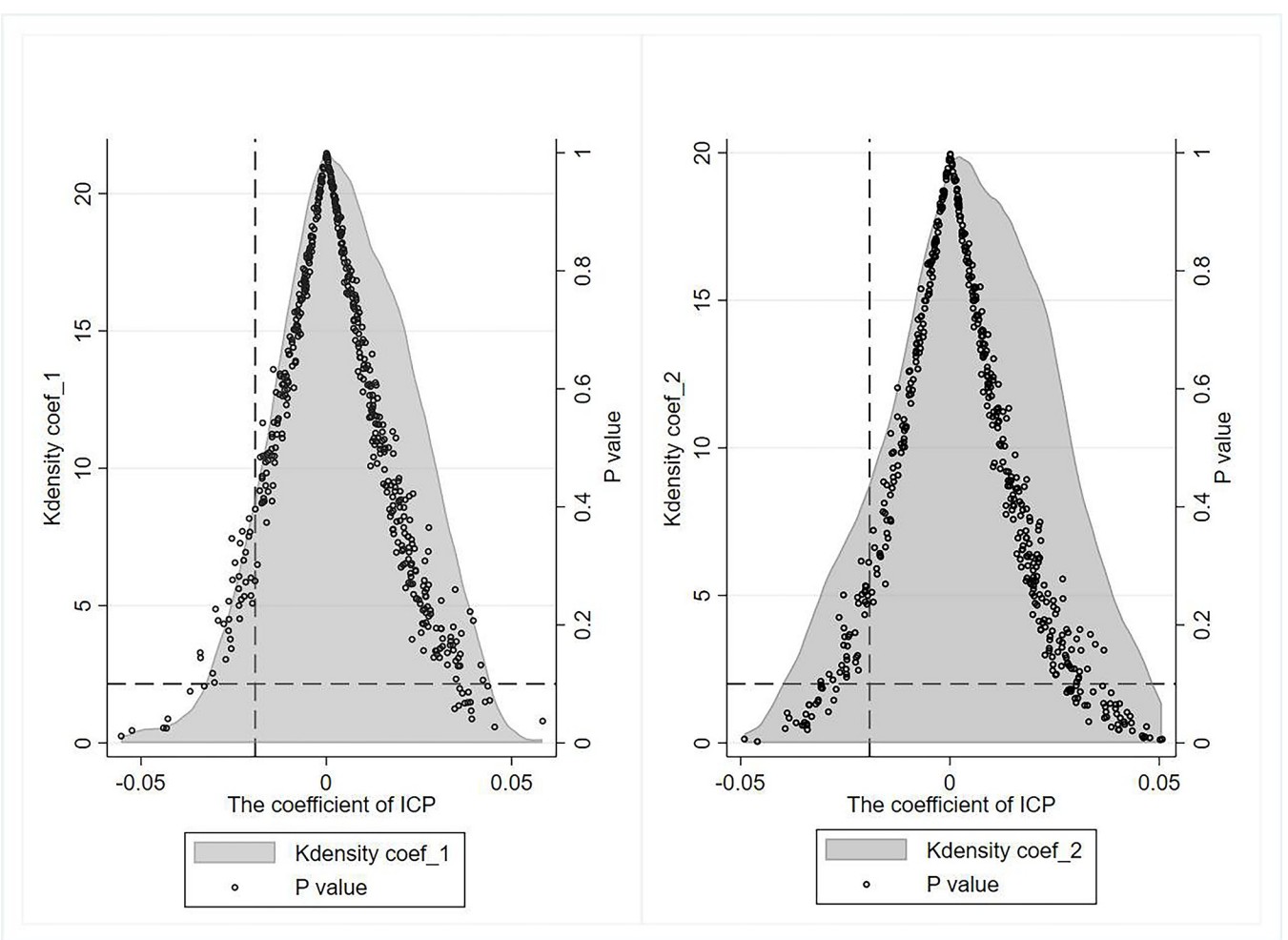

**Fig 4. Placebo test.**

## Regional heterogeneity

As economic development and innovation resource endowments differ between regions in China, innovation activities are unevenly distributed as well. We next checked whether this regional heterogeneity leads to differences in the impacts of the innovative urban pilot policy. This pilot project provides an opportunity for underdeveloped regions in the west to catch up with the east. To clarify this issue, we divided China into east, central, and west and constructed two comparative analysis groups: East and Mid-West.

Table 9 shows the regression results with $ln\_Y_{it}$ and $D.ln\_Y_{it}$ as the dependent variable to analyze the regional heterogeneity of the basic effect and convergence effect. We used 500 bootstrap samples to test whether the difference between the two sets of coefficients is different from zero and deduce the empirical p-value by estimating the distribution of the statistic. Compared with Table 9, Column 38, the coefficient of policy effect in Column 37 is numerically (0.2110) and significantly (significant at the 1% level) higher. This may be due to the uneven regional distribution of pilot policy cities and the different absorptive capacities of urban innovation. In Columns 39 and 40, the coefficient difference of the lag term $L.ln\_Y$ in Group1 is 0.112 and significant at the 1% level; that is, compared with the eastern and central regions, the urban innovation development in the western region has a higher convergence rate. Further, Columns 41 and 42 indicate that the coefficients of $L.ln\_Y \times Policy$ are significantly different. Compared with the east and mid-region (-0.0251, significant at the 1% level), they are not significant in the western region. The convergence effect of the ICP in the eastern and central regions is better than that in the western region. Thus, regional heterogeneity of the ICP on innovation convergence has been tested (including Convergence_3).

Although the western region has a faster convergence rate, ICP does not necessarily significantly promote the convergence between pilot cities in the western region. For many reasons (fewer pilot cities and poorer foundation for innovation), the influence of ICP in western China is not as significant as that in central and eastern China. Even in the western region,

**Table 9. Regional heterogeneity test (basic effect and convergence effect).**

|  | (37) | (38) | (39) | (40) | (41) | (42) |
|---|---|---|---|---|---|---|
|  | **East, Mid** | **West** | **East, Mid** | **West** | **East, Mid** | **West** |
| VARIABLES | ln_Y | | | | D.ln_Y | |
| $L.ln\_Y$ | | | -0.1400*** (-12.8800) | -0.2520*** (-12.3900) | -0.1360*** (-12.3700) | -0.2540*** (-12.4700) |
| $Policy$ | 0.2110*** (2.7300) | -0.1110 (-0.8900) | | | 0.0350 (1.6100) | -0.0849** (-2.1300) |
| $L.ln\_Y \times Policy$ | | | | | -0.0251*** (-3.7000) | 0.0130 (0.8000) |
| Coefficients difference | 0.3220** Policy | | 0.1120*** L.ln_Y | | -0.0380** L.ln_Y × Policy | |
| Control variables | yes | yes | yes | yes | yes | yes |
| Time fixed effect | yes | yes | yes | yes | yes | yes |
| Individual fixed effect | yes | yes | yes | yes | yes | yes |
| Observations | 3776 | 956 | 3556 | 901 | 3556 | 901 |
| R-squared | 0.912 | 0.912 | 0.337 | 0.451 | 0.340 | 0.353 |

Note: T-statistics are in parentheses
*** p<0.01
** p<0.05
* p<0.1.

there is heterogeneity in the effect of policy implementation among different cities. For Xi'an and Baoji, two cities in the same province in western China, the annual growth rate in the innovation index of the former was 16.4%, much higher than the 6.3% of the latter after the construction of innovation-oriented cities was implemented.

# Conclusion and suggestions

## Conclusion

As one of the determinants of a regional innovation system, the role of government in shaping regional patterns cannot be ignored. The ICP is an important exploratory policy for the government to participate in and support urban innovation development; prior studies have confirmed its positive effect on pilot cities. The conclusions of the empirical research are as follows:

1. Multiple effects of ICP
   The innovative city pilot policy not only improves the innovation level of cities (basic effect) but also promotes innovation convergence among pilot cities (convergence effect).

2. Dual character of ICP
   Although the construction of innovative cities accelerates the convergence of innovation among pilot cities, it intensifies the dispersion of innovation in the whole region (including pilot and non-pilot cities) in the short-term.

3. Time lag and spatial spillover characteristics of ICP
   Both the basic effect and the convergence effect have the characteristics of time lag and spatial spillover, but the convergence effect shows a longer time lag and a smaller range of spatial spillover than the basic effect.

4. Regional heterogeneity of ICP
   Although the western region has a faster convergence rate, the construction of innovative cities does not significantly promote the improvement of urban innovation and the convergence of innovation between cities.

## Policy implications and suggestions

Our research has several policy implications based on the above analysis and conclusions.

First, given that the pilot policy has both basic and convergent effects, on the premise of giving full consideration to urban innovation resource endowment, development environment, and regional pattern, subsequent policy formulation should continue to expand the scope of pilot cities while also planning the distribution of pilot cities in different regions and at different levels. Based on the continuous promotion of innovative city pilot projects, the increase of R&D investment and the gathering of scientific and technological talents in some regions can be realized. Through innovation agglomeration, the flow and spillover of knowledge can be promoted to achieve high-level regional innovation convergence. The conclusions also apply to other place-based innovation policy pilots, integrating the concept of coordinated development of regional innovation into policy planning to avoid further expansion of innovation differences between regions.

Second, due to the limitation of spatial spillover effect in geographical distance, the existing innovative city policy has further widened the innovation gap between the bottom cities and the top cities. Policymakers should focus on cities that are not yet within a convergent radius.

In particular, the sample for this study did exclude some cities, most of which are located in remote western regions. The regional heterogeneity test results show the imbalance of policy effects, which may be due to the difference in the accumulation degree of innovation factors. The pilot project effect presentation requires a certain knowledge base. Therefore, we may be underestimating the widening regional innovation differences. For the Chinese government, this is an important challenge for coordinating regional innovation development.

Third, government should improve the regional innovation synergy mechanism to promote the convergence of regional innovation by strengthening the innovation center's radiation effect and improving the marginal area's absorption capacity. The new modes to facilitate the cross-city flow of research funds and scientific and technological personnel should be explored. Regional collaborative innovation alliances could also be established to promote closer cross-regional integration of the innovation chain and industrial chain. By taking advantage of information and resource advantages, the innovation center could establish a cooperation platform to effectively connect the supply and demand of different cities in R&D to form a regional innovation layout with clear main functions, complementary advantages, and high-quality development.

## Supporting information

**S1 File.**
(RAR)

## Author Contributions

**Conceptualization:** Yang Haodong, Wang Chen, Wang Gaofeng.

**Data curation:** Yang Haodong.

**Funding acquisition:** Wang Gaofeng.

**Methodology:** Yang Haodong.

**Supervision:** Wang Gaofeng.

**Writing – original draft:** Yang Haodong.

**Writing – review & editing:** Wang Chen, Wang Gaofeng.

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
