## [Decision Letter · Decision Letter 0]

21 Nov 2022

PONE-D-22-22291Convergence or dispersion? The Impact of China's Innovative City Pilot Policy on Regional Innovation DifferencePLOS ONE

Dear Dr. Wang,

Thank you for submitting your manuscript to PLOS ONE. After careful consideration, we feel that it has merit but does not fully meet PLOS ONE’s publication criteria as it currently stands. Therefore, we invite you to submit a revised version of the manuscript that addresses the points raised during the review process.

We look forward to receiving your revised manuscript.

Kind regards,

Abdul Majeed

Academic Editor

PLOS ONE

Journal Requirements:

2. "PLOS requires an ORCID iD for the corresponding author in Editorial Manager on papers submitted after December 6th, 2016. Please ensure that you have an ORCID iD and that it is validated in Editorial Manager. To do this, go to ‘Update my Information’ (in the upper left-hand corner of the main menu), and click on the Fetch/Validate link next to the ORCID field. This will take you to the ORCID site and allow you to create a new iD or authenticate a pre-existing iD in Editorial Manager. Please see the following video for instructions on linking an ORCID iD to your Editorial Manager account: " ext-link-type="uri" xlink:type="simple">https://www.youtube.com/watch?v=_xcclfuvtxQ"

Reviewers' comments:

Reviewer's Responses to Questions

**Comments to the Author**

1. Is the manuscript technically sound, and do the data support the conclusions?

Reviewer #1: No

Reviewer #2: No

Reviewer #3: Yes

2. Has the statistical analysis been performed appropriately and rigorously? 

Reviewer #1: I Don't Know

Reviewer #2: No

Reviewer #3: Yes

3. Have the authors made all data underlying the findings in their manuscript fully available?

Reviewer #1: No

Reviewer #2: No

Reviewer #3: Yes

4. Is the manuscript presented in an intelligible fashion and written in standard English?

Reviewer #1: No

Reviewer #2: No

Reviewer #3: Yes

5. Review Comments to the Author

Reviewer #1: the manuscript lacks a profoud teoretical framework. the authors provide a short list of variables used for assessing the policy impact on pilot cities but they don't elaborate the indicators as such and neither the context of the pilot cities. it's not clear how the authors assess the policy impact on the citites development. there are many more factors driving cities developments than the ones listed. therefore I have strong doubts that the framework presented is suitable for such ambitious analysis.

Reviewer #2: Thank you for the opportunity to review the paper "Convergence or dispersion? The Impact of China's Innovative City Pilot Policy on Regional Innovation Difference". My main concern is about the data. The authors used data from 2003-2016, which is old. The current year of 2022 is near to end, so how can authors give policy implications based on these results? So, I suggest adding up-to-date data and revising the analysis. However, I have other comments to improve the manuscript.

• What is meant by convergence or dispersion in this study?

• Why China is chosen for the context of this study.

• The study's introduction requires some improvement to add to the study's motivation, research question, and contribution.

• The literature review section can be improved by adding more relevant studies.

• The results section should be improved by adding more discussion about the study's outcome.

• The conclusion section is long; it should be short by explaining the key findings and policy implications of the study.

Reviewer #3: Convergence or dispersion? The Impact of China's Innovative City Pilot Policy on Regional Innovation Difference

• This manuscript has many figures, so authors should keep the important ones and delete the rest.

• The introduction has more than 20 references; I think it should be reduced.

• The introduction is well written; however, the research question must be added to clarify the study's objective and what the authors want to achieve from the study's outcome.

• The literature review section can be improved by adding more UpToDate studies.

• The results section only explains the outcome of the regression models. It is important to discuss these results and why results are positive or negative with economic justification.

• The conclusion part of this research is unnecessarily long, and there are many references. So, it should be short and remove the extra details.

6. PLOS authors have the option to publish the peer review history of their article (what does this mean?). If published, this will include your full peer review and any attached files.

Reviewer #1: No

Reviewer #2: No

Reviewer #3: No

---

## [Author Response · Author response to Decision Letter 0]

17 Dec 2022

Please find responses in attachment

---

## [Decision Letter · Decision Letter 1]

13 Jan 2023

PONE-D-22-22291R1Convergence or dispersion? The impact of China's innovative city pilot policy on regional innovation differencesPLOS ONE

Dear Dr. Wang,

Thank you for submitting your manuscript to PLOS ONE. After careful consideration, we feel that it has merit but does not fully meet PLOS ONE’s publication criteria as it currently stands. Therefore, we invite you to submit a revised version of the manuscript that addresses the points raised during the review process.

We look forward to receiving your revised manuscript.

Kind regards,

Abdul Majeed

Academic Editor

PLOS ONE

Journal Requirements:

Additional Editor Comments:

The authors needs to look into following comments before acceptance. 

Reviewers' comments:

Reviewer's Responses to Questions

**Comments to the Author**

1. If the authors have adequately addressed your comments raised in a previous round of review and you feel that this manuscript is now acceptable for publication, you may indicate that here to bypass the “Comments to the Author” section, enter your conflict of interest statement in the “Confidential to Editor” section, and submit your "Accept" recommendation.

Reviewer #2: (No Response)

Reviewer #3: (No Response)

2. Is the manuscript technically sound, and do the data support the conclusions?

Reviewer #2: Yes

Reviewer #3: Yes

3. Has the statistical analysis been performed appropriately and rigorously? 

Reviewer #2: Yes

Reviewer #3: Yes

4. Have the authors made all data underlying the findings in their manuscript fully available?

Reviewer #2: Yes

Reviewer #3: Yes

5. Is the manuscript presented in an intelligible fashion and written in standard English?

Reviewer #2: Yes

Reviewer #3: Yes

6. Review Comments to the Author

Reviewer #2: The manuscript is significantly improved. However, the authors need to look into following things before this paper get published.

The authors updated the dataset however figure 1 showing the trend up to 2016 so it should be revised.

It is important to add data web links of all variables.

Table values looks same compared to previous one so please double check it.

Reviewer #3: The authors addressed the comments however I have following minor suggestion for improvement.

Research framework in figure 2 is little complicated to understand so it should be revised and make easier for general readers.

The policy implications should be enhanced little more based on the results.

The manuscript should be checked for grammar and typo error to enhance the readability.

7. PLOS authors have the option to publish the peer review history of their article (what does this mean?). If published, this will include your full peer review and any attached files.

Reviewer #2: No

Reviewer #3: No

---

## [Author Response · Author response to Decision Letter 1]

19 Jan 2023

1.Response to comment: The authors updated the dataset however figure 1 showing the trend up to 2016 so it should be revised. (Reviewer #2)

Reply: Thanks to the reminding of the reviewer, we have adjusted the content of Figure 1 and divided the innovation index and its growth rate into east, central and west regions, which is consistent with the typical characteristics of regional innovation differences in China and could also be contrasted with the regional heterogeneity test.

2.Response to comment: It is important to add data web links of all variables. (Reviewer #2)

Reply: Thanks to the reminding of the reviewer, we have uploaded the data related to the manuscript to the website: https://doi.org/10.6084/m9.figshare.21900393.v1

3.Response to comment: Table values looks same compared to previous one so please double check it.(Reviewer #2)

Reply: Thank you for the caution, we have checked the table values, the results are similar to the previous one to some extent (economic justification and significance), but the specific values have changed. For example, for an updated version, In column 1, Table 3, the coefficient of the lag term L.ln_Y is -0.1186 (absolute convergence), In column 2, the coefficient of L.ln_Y is -0.1516 (conditional convergence).

Making a contrast, in the original version, the regression coefficient of L.ln_Y are -0.0964 and -0.1153, respectively. Referring to the above process, we have checked the other tables one by one, and the results are similar.

4.Response to comment: Research framework in figure 2 is little complicated to understand so it should be revised and make easier for general readers. (Reviewer #3)

Reply: Thank you for your suggestion, the original figure involved pilot cities and non-pilot cities (affected by spillover effect and not affected by spillover effect). Considering the disadvantages of bar segmentation in figure presentation, we choose to summarize the three kinds of convergence tests and delete the figure in this manuscript:

Convergence_1: we examined whether pilot cities have a higher rate of innovation convergence than non-pilot cities.

Convergence_2: we considered the spatial spillover effects (basic effect and convergence effect) of pilot policy.

Convergence_3: we investigated potential regional heterogeneity in policy effects.

5.Response to comment: The policy implications should be enhanced little more based on the results. (Reviewer #3)

Reply: Thanks for the reminding of the reviewer, according to the conclusion, related discussion is added in policy implications.

6.Response to comment: The manuscript should be checked for grammar and typo error to enhance the readability. (Reviewer #3)

Reply: Thank you for the caution, the language of this manuscript has been polished again by professionals.

---

## [Editor Report · Decision Letter 2]

30 Jan 2023

Convergence or dispersion? The impact of China's innovative city pilot policy on regional innovation differences

PONE-D-22-22291R2

Dear Dr. Wang,

We’re pleased to inform you that your manuscript has been judged scientifically suitable for publication and will be formally accepted for publication once it meets all outstanding technical requirements.

Kind regards,

Abdul Majeed

Academic Editor

PLOS ONE
---

## [Editor Report · Acceptance letter]

27 Feb 2023

PONE-D-22-22291R2 

Convergence or dispersion? The impact of China’s innovative city pilot policy on regional innovation differences 

Dear Dr. Gaofeng:

I'm pleased to inform you that your manuscript has been deemed suitable for publication in PLOS ONE. Congratulations! Your manuscript is now with our production department. 

Kind regards, 

on behalf of

Prof. Dr. Abdul Majeed 

Academic Editor

PLOS ONE